# Solving the $2$-norm $k$-hyperplane clustering problem via multi-norm formulations

## Abstract

We tackle the 2-norm (Euclidean) $k$-Hyperplane Clustering problem ($k$-HC$_2$), which asks for finding $k$ hyperplanes that minimize the sum of squared 2-norm (Euclidean) distances between each point and its closest hyperplane. We solve the problem to global optimality via spatial branch-and-bound techniques (SBB) by strengthening a mixed integer quadratically-constrained quadratic programming formulation with constraints that arise when formulating the problem in $p$-norms with $p \neq 2$. In particular, we show that, for every (appropriately scaled) $p \in \mathbb{N} \cup \{\infty\}$, one obtains a variant of $k$-HC$_2$ whose optimal solutions yield lower bounds within a multiplicative approximation factor. We focus on the case of polyhedral norms where $p = 1, \infty$ (which admit a disjunctive-programming reformulation), and prove that strengthening the original formulation by including, on top of the original 2-norm constraints, the constraints of one of the polyhedral-norms leads to an SBB method where nonzero lower bounds are obtained in a linear (as opposed to exponential) number of SBB nodes. Experimentally, we show that our strengthened formulations lead to speedups from $\frac{1}{4}$ to 1.5 orders of magnitude, drastically improving the problem's solvability to global optimality.

## 1 Introduction

Given $m$ points $\{a_1, \ldots, a_m\}$ in $\mathbb{R}^n$, the *$k$-Hyperplane Clustering* problem, or $k$-HC$_2$, asks for identifying $k$ hyperplanes which minimize the sum of the squares of the distances between each point and the hyperplane closest to it in Euclidean (2-norm) distance.

$k$-HC$_2$ arises when relationships of *co-linearity* (in $\mathbb{R}^2$) or *co-(hyper)planarity* (in $\mathbb{R}^n$) are sought. One of the problem's most natural applications is line/surface detection in digitally-sampled images and in 3d environments Amaldi & Mattavelli (2002). More applications are found in diverse areas such medical prognosis Bradely & Mangasarian (2000), linear facility location Megiddo & Tamir (1982), discrete-time piecewise affine hybrid system identification Ferrari-Trecate et al. (2003), principal/sparse component analysis Washizawa & Cichocki (2006); He & Cichocki (2007); Tsakiris & Vidal (2017), nonlinear regression He & Qin (2010), dictionary learning Zhang et al. (2013), LiDAR data classification Kong et al. (2013), and sparse matrix representation Georgiev et al. (2007).

$k$-HC$_2$ was first introduced by Bradely & Mangasarian (2000), where it is shown that, with $k = 1$, the problem is solved by computing an eigenvalue-eigenvector pair of a suitably defined matrix built as a function of the data points. $k$-HC$_2$ is $\mathcal{NP}$-hard in any norm since fitting $m$ points in $\mathbb{R}^n$ with $k$ hyperplanes with 0 error is $\mathcal{NP}$-complete even for $n = 2$, as shown by Megiddo & Tamir (1982). To tackle $k$-HC$_2$ (without optimality guarantees) when $k \geq 2$, Bradely & Mangasarian (2000) proposed an adaptation of the popular $k$-*means* heuristic by MacQueen et al. (1967). An exact Mixed Integer Quadratically Constrained Quadratic Programming (MI-QCQP) formulation for $k$-HC$_2$ which is solvable with a spatial branch-and-bound method (SBB) is proposed by Amaldi & Coniglio (2013), together with a heuristic for larger-scale instances. Works addressing variants of $k$-HC$_2$ asking for the smallest number of hyperplanes with a distance no larger than a given $\epsilon > 0$ are found in Dhyani & Liberti (2008); Amaldi et al. (2013).

In this work, we aim at solving $k$-HC$_2$ to global optimality via a spatial branch-and-bound (SBB) method. For this, we strengthen a classical mixed-integer quadratically-constrained quadratic programming (MI-QCQP) formulation for $k$-HC$_2$ by including constraints (and variables) that arise when formulating the problem in another $p$-norm ($p \neq 2$). In particular, we show that, under mild

assumptions, the inclusion of constraints stemming from a version of $k$-HC$_2$ formulated in one of the two polyhedral norms (where $p = 1, \infty$) leads to an SBB method where a nonzero global lower bounds is obtained in a linear number of SBB nodes, as opposed to the exponential number that is necessary when the classical formulation is used. Our computational experiments reveal that our strengthened formulations lead to speedups from $\frac{1}{4}$ to 1.5 orders of magnitude, substantially improving the problem's solvability to global optimality.

## 2 PRELIMINARIES

Given a point $a \in \mathbb{R}^n$, its $p$-norm with $p \in \mathbb{N} \cup \{\infty\}$ is $\|a\|_p := \lim_{q \to p} \left( \sum_{h=1}^n |a_h|^q \right)^{1/q}$. In particular, for $p = 1, 2$, and $\infty$ we have $\|a\|_1 = \sum_{h=1}^n |a_h|^q$, $\|a\|_2 := \left( \sum_{h=1}^n |a_h|^2 \right)^{1/2}$, and $\|a\|_\infty = \max_{h \in [n]} \{ |a_h| \}$.[1] The $p$-norm point-to-hyperplane distance $d_p(a, H)$ between a point $a \in \mathbb{R}^n$ and a hyperplane $H := \{ x \in \mathbb{R}^n : x^\top w = \gamma \}$ of parameters $(w, \gamma) \in \mathbb{R}^{n+1}$ is defined as the $p$-norm distance between $a$ and the point $y \in H$ that is closest to it. Namely, $d_p(a, H) := \min_{y \in H} \|a - y\|_p$. Different arguments, including Lagrangian duality—see Mangasarian (1999), can be used to show that $d_p(a, H) = \frac{|w^\top a - \gamma|}{\|w\|_{p'}}$, where $p$ and $p'$ satisfy $\frac{1}{p} + \frac{1}{p'} = 1$.[2] For $p = 2$, $d_p(a, H)$ is called *Euclidean point-to-hyperplane* (or *orthogonal*) *distance*. In many applications, such a distance is preferred as it leads to solutions that are invariant to rotations of the data points.

In formal terms, $k$-HC$_2$ is defined as follows:

> Given $m$ points $\{a_1, \ldots, a_m\}$ in $\mathbb{R}^n$, find $k$ hyperplanes of parameters $(w_j, \gamma_j) \in \mathbb{R}^{n+1}$, $j \in [k]$, minimizing the sum of the squared 2-norm point-to-hyperplane distances $d_2(a_i, H_{j(i)})$ between each point $i$ and its closest hyperplane $j(i)$.

The, arguably, most compact nonlinear programming (NLP) formulation for $k$-HC$_2$ (with a non-smooth objective function and no binary variables) reads:

$$(k\text{-HC}_2) \qquad \min_{(w, \gamma) \in \mathbb{R}^{nk+k}} \left\{ \sum_{i=1}^m \min_{j \in [k]} \left\{ \frac{(a_i^\top w_j - \gamma_j)^2}{\|w_j\|_2^2} \right\} \right\}.$$

(Throughout the paper, we report mathematical programming formulations in brackets and optimization problems without them.)

Since $\|w_j\|_2^2 = w_j^\top w_j$, ($k$-HC$_2$) features ratios of quadratics. Besides the inner $\min$ operator, which can be easily dropped by introducing binary assignment variables (see further), such a formulation is unsuitable for most nonlinear programming solvers since the denominator vanishes when $w_j = 0$.

In particular, the distance function $d_p$ renders the problem intrinsically nonconvex regardless of the choice of $p$ (see the appendix for a proof):

**Proposition 1.** *Given a hyperplane $H := \{ x \in \mathbb{R}^n : x^\top w = \gamma \}$ and a point $a \in \mathbb{R}^n$, the function $d_p(a, H) = \frac{|w^\top a - \gamma|}{\|w\|_{p'}}$, where $\frac{1}{p} + \frac{1}{p'} = 1$, is a nonconvex function of $(w, \gamma)$ for every $p \in \mathbb{N} \cup \{\infty\}$.*

This nonconvexity motivates the adoption of SBB techniques for solving $k$-HC$_2$ to global optimality.

## 3 APPROXIMATING $k$-HC$_2$ USING DIFFERENT NORMS

In the remainder of the paper, we will work with $k$-HC$_{(p,c)}$, a generalized version of $k$-HC$_2$ which employs a $p$ norm not necessarily equal to 2 and which is parametric in a constant $c \geq 0$. Its NLP formulation, where $\frac{1}{p} + \frac{1}{p'} = 1$, reads:

$$(k\text{-HC}_{(p,c)}) \quad \min_{(w, \gamma) \in \mathbb{R}^{nk+k}} \left\{ \sum_{i=1}^m \min_{j \in [k]} \left\{ (a_i^\top w_j - \gamma_j)^2 \right\} : \|w_j\|_{p'} \geq c, j \in [k] \right\},$$

---

[1] Throughout the paper, we adopt the notation $[\xi] := 1, \ldots, \xi$ for every $\xi \in \mathbb{N}$.

[2] Two norms satisfying $\frac{1}{p} + \frac{1}{p'} = 1$ are called *dual*. The 2-norm is self dual and, assuming $\frac{1}{\infty} = 0$, the 1 and $\infty$-norms are mutually dual.

Letting, for some optimization problem $P$, $\mathrm{OPT}(P)$ be its optimal solution value, the validity of the formulation ($k$-HC$_{(p,c)}$) and the role that $c$ plays in it are shown by the following lemma (the proof is in the appendix):

**Lemma 1.** $k$-HC$_{(2,1)}$ *and* $k$-HC$_2$ *coincide. Also,* $k$-HC$_{(p,c)}$ *is quadratically homogeneous w.r.t.* $c$, *i.e.,* $\mathrm{OPT}(k\text{-HC}_{(p,c)}) = c^2 \, \mathrm{OPT}(k\text{-HC}_{(p,1)})$.

The property shown by the lemma will be useful to guide our choice of which $p$ to use for introducing additional norm constraints to the formulation of $k$-HC$_2$ (which, we recall, is the version of the problem that we aim to solve in this paper) in order to strengthen it.

$k$-HC$_{(p,c)}$ with $(p,c) \neq (2,1)$ is of interest for two reasons. First, as shown in this section, it allows us to show that, for a suitable choice of $p$ and $c$, the optimal solutions to $k$-HC$_{(p,c)}$ are approximate solutions (to within an approximation factor) of those to $k$-HC$_{(2,1)}$. Second, it allows us to show that, again for a suitable choice of $p$ and $c$, the formulations ($k$-HC$_{(p,c)}$) and ($k$-HC$_{(2,1)}$) can be intersected to obtain a *strengthened formulation* which is valid for $k$-HC$_2$ and which is also much easier to solve both in theory and practice. This is shown in Sections 4 and 5.

We remark that, while changes of norm are often used in machine learning, the presence of a dual norm in the denominator of the point-to-hyperplane distance requires, as we will show, switching between primal and dual norms and applying suitable scaling factors to the problem's constraints in a way that, to our knowledge, is specific to this problem for, in particular, the impact it has on the bounds that are obtained when solving it with an SBB method (see the analysis in Section 5).

### 3.1 THE GENERAL CASE

We show that, whichever version of $k$-HC$_{(p,c)}$ one aims to solve (be it the 2-norm one with $c = 1$ or another one), the optimal-solution value of $k$-HC$_{(q,c')}$ for *any* choice of $q$ is within an approximation factor of the optimal-solution value to $k$-HC$_{(p,c)}$, provided that $c'$ is selected correctly. This is shown by the following theorem:

**Theorem 1.** *Let* $p, q \in \mathbb{N} \cup \{\infty\}$ *and* $c > 0$. *The three positive scalars* $\alpha(p,q), \beta(p,q), \gamma(p,q)$ *which satisfy the congruence relationship*

$$\alpha(p,q)||x||_p \leq \beta(p,q)||x||_q \leq \gamma(p,q)||x||_p \qquad \forall x \in \mathbb{R}^n \tag{1}$$

*for* $p, q \in \mathbb{N} \cup \{\infty\}$ *also satisfy*

$$\frac{\alpha(p,q)^2}{\gamma(p,q)^2} \, \mathrm{OPT}(k\text{-HC}_{(p,c)}) \leq \mathrm{OPT}\left(k\text{-HC}_{(q,c\frac{\beta(p,q)}{\gamma(p,q)})}\right) \leq \mathrm{OPT}(k\text{-HC}_{(p,c)}). \tag{2}$$

The theorem shows that the optimal-solution value of $k$-HC$_{(q,c')}$ with $c' = c\frac{\beta(p,q)}{\gamma(p,q)}$ is a lower bound on the optimal solution value of $k$-HC$_{(p,c)}$ to within an approximation factor of $\frac{\alpha(p,q)^2}{\gamma(p,q)^2}$. This is important, as it shows which value to pick for $c'$ for *any* $q$-norm we may choose to obtain a relaxation of $k$-HC$_{(p,c)}$ and, in particular, one of $k$-HC$_{(2,1)}$ (which is, ultimately, the problem we aim to solve).

Let us note that the theorem can be extended to also produce an approximation of $k$-HC$_{(p,c)}$ from above to within an approximation factor. We omit the details since, in the paper, we are interested in approximations from below to build tighter relaxations suitable for an SBB method (see Section 4).

Due to the choice of $c' = c\frac{\beta(p,q)}{\gamma(p,q)}$, the feasible region of the $q$-norm constraints that corresponds to $k$-HC$_{(q,c')}$ is a relaxation of (i.e., it contains) the region that is feasible for the $p$-norm constraints of $k$-HC$_{(p,c)}$. An illustration is reported in Figure 1 for $p = 2, c = 1$ and adopting $q = 1, \infty$, for which we have $c' = 1, \frac{1}{\sqrt{n}}$ (these values are calculated in the next subsection).

### 3.2 THE CASE OF POLYHEDRAL NORMS WITH $q = 1, \infty$

We now focus on *polyhedral* norms ($q = 1, \infty$). These are of computational interest due to their greater tractability since the constraints $||w_j||_q \geq c'$, $j \in [k]$, with $q = 1, \infty$, while still non-convex, can be formulated as disjunctions over polyhedra and are thus mixed integer linear programming representable.

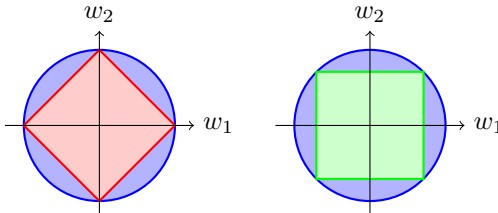

Figure 1: A depiction of the feasible region of the constraints $||w||_1 \geq 1$ and $||w||_\infty \geq \frac{1}{\sqrt{2}}$.

In light of this, we consider the following two relaxations of $k$-HC$_{(2,1)}$ (we recall that, by construction, $k$-HC$_{(p,c)}$ features the dual-norm constraints $\|w_j\|_{p'} \geq c$, $j \in [k]$, with $\frac{1}{p} + \frac{1}{p'} = 1$, for every $p \in \mathbb{N} \cup \{\infty\}$ and $c \geq 0$):

$$(k\text{-HC}_{(\infty,1)}) \quad \min_{(w,\gamma)\in\mathbb{R}^{nk+k}} \left\{ \sum_{i=1}^{m} \min_{j\in[k]} \left\{ (a_i^\top w_j - \gamma_j)^2 \right\} : \ \|w_j\|_1 \geq 1, j \in [k] \right\},$$

$$(k\text{-HC}_{(1,\frac{1}{\sqrt{n}})}) \quad \min_{(w,\gamma)\in\mathbb{R}^{nk+k}} \left\{ \sum_{i=1}^{m} \min_{j\in[k]} \left\{ (a_i^\top w_j - \gamma_j)^2 \right\} : \ \|w_j\|_\infty \geq \frac{1}{\sqrt{n}}, j \in [k] \right\}.$$

See again Figure 1 for an illustration of the feasible regions of the projection of these two problems onto the $w$ space (assuming, for simplicity, $k = 1$).

For these two variants of $k$-HC$_{(2,1)}$, Theorem 1 leads to the following result (the proof is in the appendix):

**Corollary 1.** $k$-HC$_{(\infty,1)}$ and $k$-HC$_{(1,\frac{1}{\sqrt{n}})}$ satisfy:

$$\frac{1}{n} \operatorname{OPT}(k\text{-HC}_{(2,1)}) \leq \operatorname{OPT}(k\text{-HC}_{(\infty,1)}) \leq \operatorname{OPT}(k\text{-HC}_{(2,1)})$$

$$\frac{1}{n} \operatorname{OPT}(k\text{-HC}_{(2,1)}) \leq \operatorname{OPT}(k\text{-HC}_{(1,\frac{1}{\sqrt{n}})}) \leq \operatorname{OPT}(k\text{-HC}_{(2,1)}).$$

With the first chain of inequalities, the corollary shows that solving $k$-HC$_{(\infty,1)}$, i.e., formulating $k$-HC with the constraint $\|w_j\|_1 \geq 1$ for all $j \in [k]$, leads to a relaxation to within a $\frac{1}{n}$ approximation factor. With the second one, the corollary shows that solving $k$-HC$_{(1,\frac{1}{\sqrt{n}})}$, i.e., solving the version of $k$-HC with the constraint $\|w_j\|_\infty \geq \frac{1}{\sqrt{n}}$ for all $j \in [k]$, leads to another relaxation also to within the same approximation factor $\frac{1}{n}$.

### 3.3 Multi-norm relaxation

Since both $\|w_j\|_1 \geq 1$, $j \in [k]$, and $\|w_j\|_\infty \geq \frac{1}{\sqrt{n}}$, $j \in [k]$, are relaxations of $\|w_j\|_2 \geq 1$, $j \in [k]$, a strengthened relaxation of $k$-HC$_{(2,1)}$ can be obtained by simultaneously imposing both. Such a *multi-norm* relaxation, which we refer to as $k$-HC$_{(\text{multi},1)}$, reads

$$(k\text{-HC}_{(\text{multi},1)}) \quad \min_{(w,\gamma)\in\mathbb{R}^{nk+k}} \left\{ \sum_{i=1}^{m} \min_{j\in[k]} \left\{ (a_i^\top w_j - \gamma_j)^2 \right\} : \ \begin{array}{ll} \|w_j\|_1 \geq 1, & j \in [k] \\ \|w_j\|_\infty \geq \frac{1}{\sqrt{n}}, & j \in [k] \end{array} \right\}.$$

Letting $\|w\|_{\text{multi}} := \min\{\|w\|_1, \sqrt{n}\|w\|_\infty\}$, one can see that simultaneously imposing $\|w_j\|_1 \geq 1$ and $\|w_j\|_\infty \geq \frac{1}{\sqrt{n}}$, $j \in [k]$, coincides with imposing $\|w_j\|_{\text{multi}} \geq 1, j \in [k]$. A depiction of the corresponding feasible region is reported in Figure 2.

So far, our analysis has hinged on the possibility of translating a $p'$-norm constraint into the corresponding $d_p$ distance, on which we applied Theorem 1. Analyzing the tightness of $k$-HC$_{(\text{multi},1)}$ is not as easy, though. This is because the sub-level sets of the function $\|w\|_{\text{multi}}$ are not convex and,

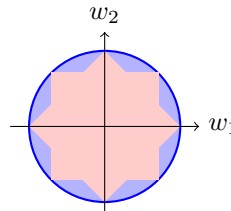

Figure 2: A depiction of the feasible region of constraint $||w||_{\text{multi}} \geq 1$.

thus, there is no $p$-norm, $p \in \mathbb{N} \cup \{\infty\}$, whose adoption leads to $k$-HC$_{(\text{multi},1)}$. In spite of this, with our analysis we will build such a function and show that it is indeed a norm.

We start with the following lemma (the proof is in the appendix), which shows what combination of point-to-hyperplane distances is maximized in $k$-HC when imposing $\min\{||w||_1, \sqrt{n}||w||_\infty\} \geq 1$:

**Lemma 2.** *Imposing* $\min\{||w||_1, \sqrt{n}||w||_\infty\} \geq 1$ *coincides with accounting for each point-t-hyperplane distance as* $\max\{d_\infty(a_i, H_j), \frac{1}{\sqrt{n}}d_1(a_i, H_j)\}$, *which translates in measuring the distance between $a_i$ and the closest point on $H_j$, call it $y$, as* $\max\{||a_i - y||_\infty, \frac{1}{\sqrt{n}}||a_i - y||_1\}$.

We now prove a second lemma (the proof is in the appendix) which shows that the function $\max\{||x||_\infty, \frac{1}{\sqrt{n}}||x||_1\}$ is a norm and which also constructs a congruence relationship for it:

**Lemma 3.** $\max\{||x||_\infty, \frac{1}{\sqrt{n}}||x||_1\}$ *is a norm and it satisfies the congruence relationship*

$$1 \Big/ \sqrt{1 + \frac{(\sqrt{n}-1)^2}{(n-1)}}||x||_2 \leq \max\{||x||_\infty, \frac{1}{\sqrt{n}}||x||_1\} \leq ||x||_2 \qquad \forall x \in \mathbb{R}^n.$$

Combining Lemma 3 with Theorem 1, we obtain the following approximation result for the multi-norm relaxation $k$-HC$_{(\text{multi},1)}$:

**Corollary 2.** $k$-HC$_{(\text{multi},1)}$ *enjoys the following approximation relationship:*

$$1 \Big/ \left(1 + \frac{(\sqrt{n}-1)^2}{(n-1)}\right) \text{OPT}(k\text{-HC}_{(2,1)}) \leq \text{OPT}(k\text{-HC}_{(\text{multi},1)}) \leq \text{OPT}(k\text{-HC}_{(2,1)}).$$

As one can see, the factor $1 \big/ \left(1 + \frac{(\sqrt{n}-1)^2}{(n-1)}\right)$ is strictly smaller than $\frac{1}{n}$, which indicates that the multi-norm relaxation $k$-HC$_{(\text{multi},1)}$ leads to, in the worst case, a bound which is at least as tight as either of those obtained individually with either of the polyhedral norms.

## 4 SOLVING STRENGTHENED FORMULATIONS OF $k$-HC$_{(2,1)}$ VIA SBB

We now focus on solving $k$-HC$_{(2,1)}$ to global optimality via SBB. We analyze the number of SBB nodes needed to compute a nonzero global lower bound when solving a basic formulation of the problem, and then prove that intersecting such a formulation with one of our relaxations involving the polyhedral norms allows for computing a nonzero global lower bounds much earlier.

### 4.1 SPATIAL BRANCH-AND-BOUND

The basic idea of the spatial branch-and-bound (SBB) method is of building a dual bound by optimizing over a convex (typically polyhedral) envelope conv($F$) of the feasible region $F$ of the problem. $F$ is then split into two sub-regions $F_1$ and $F_2$ with tighter bounds on at least a variable. This allows for constructing tighter convex envelopes of $F_1$ and $F_2$ in such a way that the optimal solution over conv($F$) is cut off due to not belonging to conv($F_1$) $\cup$ conv($F_2$). $F_1$ and $F_2$ are then recursively optimized in a classical branch-and-bound fashion.

Let us consider the case of $k$-HC$_{(2,1)}$. We assume (as done by most of the state-of-the-art solvers such as Gurobi Gurobi Optimization, LLC (2022)), that polyhedral envelopes are employed. Under

such assumption, when considering the nonlinear constraints $||w_j||_2^2 = \sum_{h=1}^{n} w_{jh}^2 \geq 1$, for $j \in [k]$, the SBB method first introduces the auxiliary variable $z_{jh}$ for each nonlinear term $w_{jh}^2$ and a corresponding defining constraint $z_{jh} = w_{jh}^2$. It then substitutes the original nonlinear constraint with $\sum_{h=1}^{n} z_{jh} \geq 1$. Each defining constraint is then relaxed into a polyhedral envelope. The point-wise minimal outer envelope of a bilinear product corresponds to the well-known McCormick envelope McCormick (1976). An illustration is reported in Figure 3 for $z_{jh} = w_{jh}^2$ with $w_{jh} \in [-1, 1]$. For a more in-depth description of SBB methods, we refer the reader to Belotti et al. (2009).

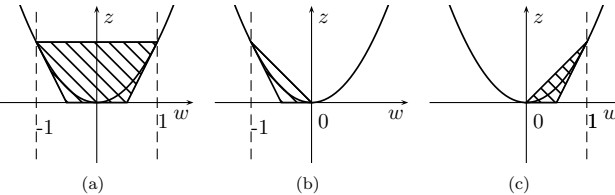

(a)           (b)           (c)

Figure 3: (a) Polyhedral envelope of $z_{jh} = w_{jh}^2$ for (a) $w_{jh} \in [-1, 1]$, (b) $w_{jh} \in [-1, 0]$, (c) $w_{jh} \in [0, 1]$. (b) and (c) are obtained after branching on $w_{jh} = 0$.

## 4.2 MATHEMATICAL PROGRAMMING FORMULATION OF $k$-HC$_{(2,1)}$

We start by considering as baseline the following classical Mixed Integer Quadratically Constrained Quadratic Programming (MI-QCQP) formulation of $k$-HC$_{(2,1)}$:

$$(k\text{-HC}_{(2,1)}) \qquad \min_{\substack{(w_j, \gamma_j) \in \mathbb{R}^{n+1} \ \forall j \in [k] \\ x_{ij} \in \{0,1\} \ \forall (i,j) \in [m] \times [k]; d \in \mathbb{R}_+^m}} \sum_{i=1}^{m} d_i^2 : \left\{ \begin{array}{ll} \sum_{j=1}^{n} x_{ij} = 1 & \forall i \in [m] \\ ||w_j||_2 \geq 1 & \forall j \in [k] \\ d_i \geq w_j^T a_i - \gamma_j - d^U(1 - x_{ij}) & \forall i \in [m], j \in [k] \\ d_i \geq -w_j^T a_i + \gamma_j - d^U(1 - x_{ij}) & \forall i \in [m], j \in [k] \end{array} \right\} .$$

In it, $x_{ij} \in \{0, 1\}$ takes value 1 if and only if $a_i$ is assigned to the hyperplane of index $j \in [k]$; $d_i$ is the distance between $a_i$ and the hyperplane of index $j \in [k]$; $d^U$ is an upper bound on the largest distance between any point $a_i$ and hyperplane of index $j \in [k]$. The only nonconvexity of the formulation is due to the 2-norm constraints. W.l.o.g., we assume $a_i \geq 0$ for all $i \in [m]$ (as this can be easily obtained by translating the dataset).

The following bounds on the variables can be included. We let $d^U := ||b \ e||_2$, where $e$ is the all-one vector and $b$ is the length of the edge of the smallest hypercube that contains $\{a_1, \ldots, a_m\}$. Since $||w_j||_2 = 1$ holds in any optimal solution and $\max\{||w_j||_\infty : ||w_j||_2 = 1\} = 1$, we impose $||w_j||_\infty \leq 1$ via $-e \leq w_j \leq e$, $j \in [k]$. These bounds imply $-nb - d^U \leq \gamma_j \leq nb + d^U$, $j \in [k]$.

Since the point-to-hyperplane distance is symmetric, given any solution to $k$-HC$_{(2,1)}$, an equivalent one can be obtained by changing the sign of $w_j$ for some $j \in [k]$. To remove such a symmetry (symmetries are known to be a hindrance when solving mathematical programming problems to optimality via methods based on (spatial) branch-and-bound), we impose $w_j$ to belong to an arbitrary half-space of $\mathbb{R}^n$ for each $j \in [n]$ by imposing $w_{j1} \geq 0, j \in [k]$, where $w_{j1}$ is the first component of $w_j$. In this way, any solution that is obtainable by changing the sign of a component of one of the vectors $w_j$ becomes infeasible (due to being obtained from the previous one by reflection of $w_j$ over the hyperplane defining the halfspace that we selected), thus breaking the symmetry. [3]

## 4.3 SOLVING THE FORMULATION $(k\text{-HC}_{(2,1)})$ VIA SBB

Let us now analyze the behavior of an SBB method when solving the classical formulation $(k\text{-HC}_{(2,1)})$. Since the projection onto the $w$ space of the feasible region of $k$-HC$_{(2,1)}$ is nonconvex

---

[3]In all our formulations, we partially remove the symmetry on $x_{ij}$, $i \in [m], j \in [k]$, that is induced by the assignment constraints by imposing $x_{ij} = 0$ for all $i, j \in [m] \times [k]$ with $i < j$. This reduces the number of 0-1 variables by $\sum_{h=1}^{k-1} \frac{(k-1)k}{2}$.

and its complement is symmetric about the origin, any SBB method based on convex envelopes will necessarily convexify the infeasible region, thus making the trivial solution $w_j = 0, j \in [k]$, feasible. This leads to a bound as weak as possible due to the fact that the objective function is the sum of squares $\sum_{i=1}^{m} d_i^2 \geq 0$ and, with $(w_j, \gamma_j) = 0, j \in [k]$, we obtain $\sum_{i=1}^{m} d_i^2 = 0$.

Let us make the following assumption, which holds in many SBB codes Belotti et al. (2009):

**Assumption 1.** *Assume that, when spatially branching on variables with a symmetric domain (due to the bounds we included, the domain of $w_{jh}, j \in [k], h \in [n]$, is symmetric), branching takes place on the mid point of the domain.*

Crucially, under such an assumption the geometry of the feasible region of $k$-HC$_{(2,1)}$ makes it so that the number of branching operations that are needed to make the 0 solution infeasible (and, thus, compute a nonzero global lower bound) is exponentially large:

**Proposition 2.** *Under Assumption 1, when solving $k$-HC$_{(2,1)}$ a nonzero lower bound is obtained only after generating $\Omega(2^{k(n-1)})$ nodes.*

### 4.4 STRENGTHENED FORMULATIONS

We now construct valid formulations for $k$-HC$_2$ which are strengthened by featuring not only the 2-norm constraints but also a collection of polyhedral-norm constraints. Building on the relaxations we constructed before, we introduce the following three strengthened formulations:

$$(k\text{-HC}_{(2,1),(\infty,1)}) \quad \min_{(w,\gamma)\in\mathbb{R}^{nk+k}]} \left\{ \sum_{i=1}^{m} \min_{j\in[k]} \left\{ (a_i^\top w_j - \gamma_j)^2 \right\} : \begin{array}{ll} \|w_j\|_2 \geq 1, & j \in [k] \\ \|w_j\|_1 \geq 1, & j \in [k] \end{array} \right\},$$

$$(k\text{-HC}_{(2,1),(1,\frac{1}{\sqrt{n}})}) \quad \min_{(w,\gamma)\in\mathbb{R}^{nk+k}} \left\{ \sum_{i=1}^{m} \min_{j\in[k]} \left\{ (a_i^\top w_j - \gamma_j)^2 \right\} : \begin{array}{ll} \|w_j\|_2 \geq 1, & j \in [k] \\ \|w_j\|_\infty \geq \frac{1}{\sqrt{n}}, & j \in [k] \end{array} \right\},$$

$$(k\text{-HC}_{(2,1),(\text{multi},1)}) \quad \min_{(w,\gamma)\in\mathbb{R}^{nk+k}} \left\{ \sum_{i=1}^{m} \min_{j\in[k]} \left\{ (a_i^\top w_j - \gamma_j)^2 \right\} : \begin{array}{ll} \|w_j\|_2 \geq 1, & j \in [k] \\ \|w_j\|_1 \geq 1, & j \in [k] \\ \|w_j\|_\infty \geq \frac{1}{\sqrt{n}}, & j \in [k] \end{array} \right\}.$$

Before analyzing the number of branching operations needed to achieve a nonzero lower bound with these relaxations, we report the Mixed Integer Linear Programming (MILP) formulations by which we formulate the polyhedral-norm constraints.

We formulate the constraints $\|w_j\|_1 \geq 1, j \in [k]$, via the following absolute-value reformulation:

$$w_{jh}^+ - w_{jh}^- = w_{jh} \qquad\qquad h \in [n] \tag{3a}$$

$$w_{jh}^+ \leq s_{jh} \qquad\qquad h \in [n] \tag{3b}$$

$$w_{jh}^- \leq (1 - s_{jh}) \qquad\qquad h \in [n] \tag{3c}$$

$$\sum_{h=1}^{n} (w_{jh}^+ + w_{jh}^-) \geq 1 \tag{3d}$$

$$0 \leq w_{jh}^+, w_{jh}^- \leq 1 \qquad\qquad h \in [n] \tag{3e}$$

$$s_{jh} \in \{0,1\}^n \qquad\qquad h \in [n]. \tag{3f}$$

The binary variable $s_{jh}$ denotes the sign of the $h$-th component of $w_j$. Consider a component $w_{jh}$ of index $h$ of $w_j$. Due to Constraints (3a)–(3c), if $w_{jh} > 0$, then $w_{jh}^+ > 0$ (with $w_{jh}^+ = w_{jh}$ and $w_{jh}^- = 0$) and $s_{jh} = 1$. Otherwise, if $w_{jh} < 0$, then $w_{jh}^- > 0$ (with $w_{jh}^+ = 0$ and $w_{jh}^- = -w_{jh}$) and $s_{jh} = 0$. Since $w_j^+$ and $w_j^-$ are component-wise complementary thanks to Constraints (3b)–(3c), we deduce that $w_j^+ + w_j^- = |w_j|$ holds. Thus, Constraint (3d) guarantees $\|w_j\|_1 \geq 1$.

When these constraints are imposed, we break symmetry as mentioned before by imposing $w_{j1} \geq 0$, $j \in [k]$. This leads to $s_{j1} = 1$ and $w_{j1}^- = 0$, thanks to which Constraint (3d) becomes $w_{j1} + \sum_{h=2}^{n} (w_{jh}^+ + w_{jh}^-) \geq 1$.

We formulate the constraints $\|w_j\|_\infty \geq \frac{1}{\sqrt{n}}$, $j \in [k]$, i.e., $\max_{h \in [n]}\{|w_{jh}|\} \geq \frac{1}{\sqrt{n}}$, $j \in [k]$, as the disjunction $\bigvee_{h=1}^n \left( w_{jh} \leq -\sqrt{n} \vee w_{jh} \geq \frac{1}{\sqrt{n}} \right), j \in [k]$. Differently from the previous cases, in this case we break symmetry by (w.l.o.g.) always selecting $w_{jh} \geq \frac{1}{\sqrt{n}}$ from each elementary disjunction $w_{jh} \leq -\frac{1}{\sqrt{n}} \vee w_{jh} \geq \frac{1}{\sqrt{n}}$. This translates into considering the restricted disjunction $\bigvee_{h=1}^n w_{jh} \geq \frac{1}{\sqrt{n}}$, $j \in [k]$. For each $j \in [k]$, we restate the resulting disjunctive set via the following MILP formulation:

$$w_{jh} \geq \frac{1}{\sqrt{n}} \left( \frac{1}{\sqrt{n}} - 2(1 - u_{jh}) \right) \qquad\qquad h \in [n] \tag{4a}$$

$$\sum_{h=1}^n u_{jh} = 1 \tag{4b}$$

$$u_{jh} \in \{0, 1\} \qquad\qquad h \in [n]. \tag{4c}$$

Due to Constraint (4a), if $u_{jh} = 1$ holds for some $h \in [n]$, then $w_{jh} \geq \frac{1}{\sqrt{n}}$ holds (the constraint is inactive if $u_{jh} = 0$, and reads $w_{jh} \geq -\frac{1}{\sqrt{n}}$). Constraint (4b) imposes that exactly a component of $u_j = (u_{j1}, \ldots, u_{jn})$ be equal to 1.

When imposing multiple norm constraints at once, we only have to pay attention to the way symmetry is prevented, as the symmetry-breaking constraint $w_{j1} \geq 0$ we introduced for the constraints $\|w_j\|_2 \geq 1$, $j \in [k]$, and $\|w_j\|_1 \geq 1$, $j \in [k]$, is not compatible with the one-sided disjunction we considered for $\|w_j\|_\infty \geq \frac{1}{\sqrt{n}}$, $j \in [k]$, and imposing both would not lead to an over-restriction. Whenever the $\|w_j\|_\infty \geq \frac{1}{\sqrt{n}}$ constraints are imposed, we sort the issue by dropping the symmetry-breaking constraints $w_{jh} \geq 0$, $j \in [k]$.

## 4.5 SOLVING THE STRENGTHENED FORMULATIONS VIA SBB

We extend the analysis in Proposition 2 to the strengthened formulations with the following two propositions (their proofs of both are contained in the appendix):

**Proposition 3.** *Assume that the constraint $\|w_j\|_1 \geq 1$, $j \in [k]$, is imposed and that branching takes place on the $s_{jh}$ variables first. Then, a nonzero global lower bound is calculated after generating $\Theta(2^{k(n-1)})$ nodes; after this, no further branching on $w$ takes place.*

**Proposition 4.** *Assume that $\|w_j\|_\infty \geq \frac{1}{\sqrt{n}}$, $j \in [k]$, is imposed and that branching takes place on the $u_{jh}$ variables first. Then, $O(nk)$ nodes suffice to obtain a nonzero lower bound; after this, no further branching on $w$ takes place.*

Propositions 3 and 4 show the crucial advantages of strengthening formulation ($k$-HC$_{(2,1)}$) via the two (scaled) polyhedral-norm constraints we considered. Proposition 3 indicates that, if the $\|w_j\|_1 \geq 1, j \in [k]$, constraints are imposed and branching takes places on the 0-1 variables of such norm constraints, in a complete SBB tree of depth $\Theta(2^{k(n-1)})$ the polyhedral-norm constraint is satisfied in *every* leaf node. While this number can be large, this is in stark contrast to the 2-norm case, where the same number of branching operations only suffices to obtain the first nonzero global lower bound, and the number of branchings needed to completely describe the feasible region of the problem in the $w$ space depends on the solver's feasibility tolerance (since, for each $j \in [k]$, the complement of the feasible region is a sphere). More interestingly, Proposition 4 shows that, when the $\|w_j\|_\infty \geq \frac{1}{\sqrt{n}}$, $j \in [k]$, constraints are imposed and branching takes places on their 0-1 variables, the size the SBB tree is extremely small—only polynomial in $k$ and $n$. The difference between the two results is obviously due to the geometry of the 1- and $\infty$-norm balls, since the former has $2^n$ facets while the latter only $2n$.

When included in a formulation for $k$-HC$_2$ on top of the $\|w_j\|_2 \geq 1, j \in [k]$, constraints, the polyhedral-norm constraints accelerate the computation of a nonzero global lower bound, thus leading to more pruning and, overall, a faster SBB method. This is better shown in the next section.

## 5 COMPUTATIONAL RESULTS

We assess the effectiveness of our strengthened formulations with Gurobi 9.5's SBB using 12 threads on a 2.6GHz Intel Core i7-9750H equipped with 32 GB RAM, with a total time limit across the 12 cores of 168,000 seconds (46 hours).

We consider two testbeds: `Low-dim` and `High-dim`. `Low-dim` contains 84 instances with $m = 10, \ldots, 30$, $n = 2, 3$, and $k = 2, 3$. These instances are a superset of the 24 instances tackled with SBB techniques in Amaldi & Coniglio (2013). `High-dim` contains 43 instances with $m = 10, \ldots, 17$, $n = 2, 3, 4, 5$, and $k = 2, 3, 4, 5$. Both datasets are generated by randomly choosing $(w_j, \gamma_j)$, $j \in [k]$, with a uniform distribution in $[-1, 1]$ and distributing uniformly at random the $m$ points such that each of them belongs (with 0 distance) to a hyperplane. Then, an orthogonal deviation from the corresponding hyperplane is added to each point by sampling a Gaussian distribution with 0 mean and a variance that is selected, for each hyperplane, uniformly at random in $[0.7 \cdot 0.003, 0.003]$.

A selection of the results is reported in Tables 1 and 2. Let us focus first on the `Low-dim` testbed. With the three strengthened formulations $(k\text{-HC}_{(2,1),(1,\frac{1}{\sqrt{n}})})$, $(k\text{-HC}_{(2,1),(\infty,1)})$, and $(k\text{-HC}_{(2,1),(\text{multi},1)})$, 11 instances that are not solved in over 46 hours with the classical formulation $(k\text{-HC}_{(2,1)})$ are solved in under 2 hours. With the strengthened formulations, the 60 instances that are also solved with the classical formulation are solved, respectively, 22, 21, and 22 times faster (on geometric average). When compared to the classical formulation across the whole `Low-dim` testbed, the strengthened formulations lead to an improvement of the computing times by, respectively, 53, 52, and 67 times (on geometric average).

Interestingly, our results on the `Low-dim` testbed prove that all the heuristic solutions found in Amaldi & Coniglio (2013) on the 24 instances therein considered (those with $m = 10, 14, 18, 22, 26, 30$) are optimal.

Let us turn now to the `High-dim` testbed. On it, with the best-performing of the strengthened formulations we manage to solve 10 more instances then with the classical formulation. Compared to the classical formulation across the whole `High-dim` testbed, the strengthened formulations $(k\text{-HC}_{(2,1),(1,\frac{1}{\sqrt{n}})})$, $(k\text{-HC}_{(2,1),(\infty,1)})$, $(k\text{-HC}_{(2,1),(\text{multi},1)})$ lead to an improvement of the computing times by, respectively, 4.4, 2.5, and 1.9 times (on geometric average).

While still extremely large, these speedup factors are about 1 to 1.5 orders of magnitude smaller than those we obtained on the `Low-dim` testbed. In particular, the speedup obtained with $(k\text{-HC}_{(2,1),(\text{multi},1)})$ is smaller than the ones obtained with $(k\text{-HC}_{(2,1),(\infty,1)})$ and $(k\text{-HC}_{(2,1),(1,\frac{1}{\sqrt{n}})})$. Such a behavior is well explained by the results of Propositions 3 and 4: As $n$ and $k$ increase, the difference between the exponential lower bound (on the number of nodes required to obtain a nonzero global lower bound) in the first proposition and the polynomial one in the second one becomes larger and larger. Thus, any branching operations taking place on the $\|w_j\|_1 \geq 1$, $j \in [k]$, constraints have a much smaller impact on the bound than those taking place on the $\|w_j\|_\infty \geq \frac{1}{\sqrt{n}}$, $j \in [k]$, which explains the superior performance of $(k\text{-HC}_{(2,1),(\infty,1)})$.

## 6 CONCLUDING REMARKS

We have focused on solving the 2-norm $k$-Hyperplane Clustering problem with spatial branch-and-bound (SBB) techniques by strengthening the classical formulation with constraints that arise from (scaled) $p$-norm formulations of the problem, with $p \neq 2$. Focusing on the 1- and $\infty$-norms, we have theoretically shown that including the constraints stemming from the 1-norm version of the problem leads to computing nonzero lower bounds in a linear (rather than exponential) number of SBB nodes. Our experimental results show speedups from $\frac{1}{4}$ to 1.5 orders of magnitude, substantially improving the problem's solvability to global optimality.

Future works include addressing the problem's combinatorial (assignment) aspect, which, as the number of data points increases, may become a limiting factor when solving $k\text{-HC}_2$ to global optimality, and extending our techniques to other problems featuring nonconvex $p$-norm constraints.

Table 1: Results on the `LowDim` dataset (suboptimal values are in italics).

| | | | $(k\text{-HC}_{(2,1)})$ | | $(k\text{-HC}_{(2,1),(\infty,1)})$ | | $(k\text{-HC}_{(2,1),(1,\frac{1}{\sqrt{n}})})$ | | $(k\text{-HC}_{(\text{multi},1)})$ | |
|---|---|---|---|---|---|---|---|---|---|---|
| m | n | k | obj | time | obj | time | obj | time | obj | time |
| 10 | 2 | 2 | 0.3 | 0.3 | 0.3 | 0.2 | 0.3 | 0.2 | 0.3 | 0.2 |
| 10 | 2 | 3 | 0.5 | 0.7 | 0.5 | 1.0 | 0.5 | 0.8 | 0.5 | 1.0 |
| 14 | 2 | 2 | 8.5 | 1.6 | 8.5 | 0.6 | 8.5 | 0.2 | 8.5 | 0.3 |
| 14 | 2 | 3 | 0.8 | 31.9 | 0.8 | 4.4 | 0.8 | 3.4 | 0.8 | 5.4 |
| 18 | 2 | 2 | 3.4 | 13.9 | 3.4 | 0.4 | 3.4 | 0.4 | 3.4 | 0.7 |
| 18 | 2 | 3 | 0.7 | 488.9 | 0.7 | 3.9 | 0.7 | 4.4 | 0.7 | 4.6 |
| 22 | 2 | 2 | 9.7 | 179.2 | 9.7 | 1.7 | 9.7 | 1.4 | 9.7 | 0.9 |
| 22 | 2 | 3 | 2.4 | 2213.3 | 2.4 | 11.2 | 2.4 | 11.2 | 2.4 | 9.8 |
| 25 | 2 | 2 | 8.2 | 28.9 | 8.2 | 0.6 | 8.2 | 0.4 | 8.2 | 1.4 |
| 25 | 2 | 3 | 2.7 | 168000.0 | 2.7 | 936.6 | 2.7 | 96.1 | 2.7 | 221.0 |
| 26 | 2 | 2 | - | 168000.0 | 5.8 | 6.2 | 5.8 | 10.4 | 5.8 | 2.2 |
| 26 | 2 | 3 | - | 168000.0 | 3.4 | 39.2 | 3.4 | 56.6 | 3.4 | 28.3 |
| 27 | 2 | 2 | - | 168000.0 | 5.1 | 0.7 | 5.1 | 2.6 | 5.1 | 0.8 |
| 27 | 2 | 3 | - | 168000.0 | 3.3 | 1678.4 | 3.3 | 2687.7 | 3.3 | 238.6 |
| 28 | 2 | 2 | - | 168000.0 | 11.7 | 8.6 | 11.7 | 6.3 | 11.7 | 1.8 |
| 28 | 2 | 3 | - | 168000.0 | 3.6 | 293.1 | 3.6 | 471.3 | 3.6 | 153.5 |
| 29 | 2 | 2 | - | 168000.0 | 7.1 | 0.8 | 7.1 | 0.3 | 7.1 | 0.8 |
| 29 | 2 | 3 | - | 168000.0 | 7.1 | 7694.9 | 7.1 | 6029.0 | 7.1 | 1476.4 |
| 30 | 2 | 2 | - | 168000.0 | 9.1 | 10.4 | 9.1 | 38.5 | 9.1 | 1.6 |
| 30 | 2 | 3 | - | 168000.0 | 3.4 | 172.9 | 3.4 | 191.2 | 3.4 | 44.3 |
| 10 | 3 | 2 | 0.9 | 1.1 | 0.9 | 0.4 | 0.9 | 1.0 | 0.9 | 0.9 |
| 10 | 3 | 3 | 0.0 | 30.2 | 0.0 | 32.6 | 0.0 | 31.9 | 0.0 | 41.9 |
| 14 | 3 | 2 | 0.7 | 8.4 | 0.7 | 0.8 | 0.7 | 0.8 | 0.7 | 1.4 |
| 14 | 3 | 3 | 0.1 | 206.4 | 0.1 | 29.7 | 0.1 | 25.5 | 0.1 | 49.7 |
| 18 | 3 | 2 | 0.7 | 160.6 | 0.7 | 3.7 | 0.7 | 7.8 | 0.7 | 4.5 |
| 18 | 3 | 3 | 0.4 | 2234.9 | 0.4 | 93.4 | 0.4 | 91.6 | 0.4 | 157.9 |
| 22 | 3 | 2 | 4.3 | 625.0 | 4.3 | 15.6 | 4.3 | 11.3 | 4.3 | 10.8 |
| 22 | 3 | 3 | 1.3 | 135362.9 | 1.3 | 1089.5 | 1.3 | 638.2 | 1.3 | 1243.7 |
| 23 | 3 | 2 | 0.9 | 6459.4 | 0.9 | 8.1 | 0.9 | 45.5 | 0.9 | 10.1 |
| 24 | 3 | 2 | 6.9 | 18049.6 | 6.9 | 66.3 | 6.9 | 474.7 | 6.9 | 34.5 |
| 24 | 3 | 3 | *1.7* | 168000.0 | 1.5 | 2470.6 | 1.5 | 2716.7 | 7.9 | 3817.0 |
| 25 | 3 | 2 | 5.7 | 22886.9 | 5.7 | 70.7 | 5.7 | 28.1 | 8.9 | 14.2 |
| 25 | 3 | 3 | *1.3* | 168000.0 | 1.3 | 1952.3 | 1.3 | 5060.3 | 9.9 | 2885.1 |
| 26 | 3 | 2 | - | 168000.0 | 4.5 | 6.3 | 4.5 | 4.7 | 10.9 | 4.4 |
| 26 | 3 | 3 | - | 168000.0 | 1.3 | 5937.9 | 1.3 | 4345.7 | 11.9 | 2300.2 |
| 27 | 3 | 2 | - | 168000.0 | 3.4 | 215.1 | 3.4 | 1274.8 | 12.9 | 58.5 |
| 27 | 3 | 3 | - | 168000.0 | 2.9 | 52548.9 | 2.9 | 65949.3 | 13.9 | 35206.1 |
| 28 | 3 | 2 | - | 168000.0 | 3.6 | 31.1 | 3.6 | 1.7 | 14.9 | 10.2 |
| 28 | 3 | 3 | - | 168000.0 | 1.4 | 4234.9 | 1.4 | 74560.6 | 15.9 | 4180.9 |
| 29 | 3 | 2 | - | 168000.0 | 8.1 | 143.5 | 8.1 | 34.0 | 16.9 | 12.5 |
| 29 | 3 | 3 | - | 168000.0 | *4.9* | 168000.0 | 4.9 | 168000.0 | *17.9* | 168000.0 |
| 30 | 3 | 2 | - | 168000.0 | 2.5 | 8083.1 | 2.5 | 168000.0 | 18.9 | 3014.8 |
| 30 | 3 | 3 | - | 168000.0 | 3.2 | 23488.8 | 3.2 | 168000.0 | 19.9 | 6541.5 |

Table 2: Results on the `HighDim` dataset (suboptimal values are in italics).

| | | | $(k\text{-HC}_{(2,1)})$ | | $(k\text{-HC}_{(2,1),(\infty,1)})$ | | $(k\text{-HC}_{(2,1),(1,\frac{1}{\sqrt{n}})})$ | | $(k\text{-HC}_{(\text{multi},1)})$ | |
|---|---|---|---|---|---|---|---|---|---|---|
| m | n | k | obj | time | obj | time | obj | time | obj | time |
| 10 | 2 | 4 | 0.0 | 8.3 | 0.0 | 2.4 | 0.0 | 1.8 | 0.0 | 6.8 |
| 10 | 4 | 2 | 0.0 | 4.9 | 0.0 | 0.8 | 0.0 | 6.1 | 0.0 | 3.9 |
| 11 | 2 | 4 | 0.1 | 21.9 | 0.1 | 9.8 | 0.1 | 5.9 | 0.1 | 17.7 |
| 11 | 2 | 5 | 0.0 | 1264.3 | 0.0 | 392.8 | 0.0 | 300.2 | 0.0 | 2689.7 |
| 11 | 4 | 2 | 0.0 | 5.4 | 0.0 | 1.6 | 0.0 | 1.6 | 0.0 | 2.1 |
| 12 | 2 | 4 | 0.1 | 79.4 | 0.1 | 17.0 | 0.1 | 8.1 | 0.1 | 30.5 |
| 12 | 2 | 5 | 0.0 | 425.6 | 0.0 | 160.4 | 0.0 | 56.8 | 0.0 | 282.8 |
| 12 | 4 | 2 | 0.1 | 17.3 | 0.1 | 1.2 | 0.1 | 7.7 | 0.1 | 10.1 |
| 12 | 5 | 2 | 0.0 | 29.3 | 0.0 | 14.4 | 0.0 | 16.4 | 0.0 | 26.1 |
| 13 | 2 | 4 | 0.1 | 238.2 | 0.1 | 19.4 | 0.1 | 14.6 | 0.1 | 38.4 |
| 13 | 2 | 5 | 0.0 | 935.1 | 0.0 | 127.1 | 0.0 | 55.8 | 0.0 | 170.7 |
| 13 | 3 | 4 | 0.0 | 4143.7 | 0.0 | 7567.6 | - | 168000.0 | - | 168000.0 |
| 13 | 4 | 2 | 0.1 | 13.0 | 0.1 | 6.5 | 0.1 | 2.1 | 0.1 | 9.3 |
| 13 | 4 | 3 | 0.0 | 948.7 | 0.0 | 567.1 | 0.0 | 712.6 | 0.0 | 4625.7 |
| 13 | 5 | 2 | 0.1 | 47.0 | 0.1 | 11.1 | 0.1 | 19.8 | 0.1 | 28.3 |
| 14 | 2 | 4 | 0.2 | 683.1 | 0.2 | 22.4 | 0.2 | 12.2 | 0.2 | 55.8 |
| 14 | 2 | 5 | 0.0 | 6526.6 | 0.0 | 628.6 | 0.0 | 211.9 | 0.0 | 586.0 |
| 14 | 3 | 4 | - | 168000.0 | 0.0 | 2757.6 | 0.0 | 2784.8 | 0.0 | 7540.2 |
| 14 | 4 | 2 | 0.5 | 58.5 | 0.5 | 2.2 | 0.5 | 7.0 | 0.5 | 9.6 |
| 14 | 4 | 3 | 0.0 | 1447.5 | 0.0 | 687.9 | 0.0 | 890.5 | 0.0 | 6906.7 |
| 14 | 5 | 2 | 0.1 | 120.1 | 0.1 | 13.8 | 0.1 | 21.5 | 0.1 | 36.3 |
| 15 | 2 | 4 | 0.3 | 1350.6 | 0.3 | 32.9 | 0.3 | 23.4 | 0.3 | 54.4 |
| 15 | 2 | 5 | 0.0 | 5854.2 | 0.0 | 320.5 | 0.0 | 92.9 | 0.0 | 445.3 |
| 15 | 3 | 4 | - | 168000.0 | 0.0 | 2760.8 | 0.0 | 1772.1 | - | 168000.0 |
| 15 | 4 | 2 | 0.6 | 37.5 | 0.6 | 5.8 | 0.6 | 8.4 | 0.6 | 9.2 |
| 15 | 4 | 3 | 0.0 | 3803.0 | 0.0 | 515.6 | 0.0 | 439.4 | 0.0 | 2208.8 |
| 15 | 5 | 2 | 0.1 | 98.1 | 0.1 | 13.5 | 0.1 | 40.7 | 0.1 | 35.0 |
| 16 | 2 | 4 | 0.2 | 5827.2 | 0.2 | 119.6 | 0.2 | 28.9 | 0.2 | 67.3 |
| 16 | 2 | 5 | - | 168000.0 | 0.0 | 582.6 | 0.0 | 346.6 | 0.0 | 781.9 |
| 16 | 3 | 4 | - | 168000.0 | 0.0 | 4586.5 | 0.0 | 2407.2 | - | 168000.0 |
| 16 | 3 | 5 | - | 168000.0 | - | 168000.0 | - | 168000.0 | - | 168000.0 |
| 16 | 4 | 2 | 1.1 | 179.0 | 1.1 | 12.9 | 1.1 | 15.0 | 1.1 | 12.1 |
| 16 | 4 | 3 | 0.0 | 5144.2 | 0.0 | 554.5 | 0.0 | 601.1 | 0.0 | 2507.3 |
| 16 | 5 | 2 | 0.8 | 444.9 | 0.8 | 28.5 | 0.8 | 43.2 | 0.8 | 60.8 |
| 17 | 2 | 4 | 0.2 | 168000.0 | 0.2 | 37.1 | 0.2 | 42.1 | 0.2 | 69.2 |
| 17 | 2 | 5 | 0.1 | 168000.0 | 0.1 | 1452.3 | 0.1 | 999.4 | 0.1 | 1517.1 |
| 17 | 3 | 4 | - | 168000.0 | 0.0 | 4970.5 | 0.0 | 2553.9 | - | 168000.0 |
| 17 | 3 | 5 | - | 168000.0 | - | 168000.0 | - | 168000.0 | - | 168000.0 |
| 17 | 4 | 2 | 0.5 | 175.7 | 0.5 | 9.8 | 0.5 | 10.6 | 0.5 | 9.8 |
| 17 | 4 | 3 | - | 168000.0 | 0.0 | 904.1 | 0.0 | 967.5 | 0.0 | 3679.0 |
| 17 | 4 | 4 | - | 168000.0 | 0.0 | 8218.2 | 1.4 | 97.4 | 0.0 | 8104.9 |
| 17 | 5 | 2 | 1.4 | 1092.7 | 1.4 | 87.0 | 1.4 | 97.4 | 1.4 | 101.0 |
| 17 | 5 | 3 | - | 168000.0 | 0.0 | 8116.4 | 0.0 | 8082.4 | 0.0 | 7910.9 |

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
