## A   CODE REPOSITORY AND LICENSING

The code developed for this work is available at `https://anonymous.4open.science/r/norms-5F23`.

## B   LIST OF OUR THEORETICAL RESULTS WITH THE CORRESPONDING PROOFS

**Proposition 1.** *Given a hyperplane $H := \{x \in \mathbb{R}^n : x^\top w = \gamma\}$ and a point $a \in \mathbb{R}^n$, the function $d_p(a, H) = \frac{|w^\top a - \gamma|}{\|w\|_{p'}}$, where $\frac{1}{p} + \frac{1}{p'} = 1$, is a nonconvex function of $(w, \gamma)$ for every $p \in \mathbb{N} \cup \{\infty\}$.*

*Proof.* By definition, $\frac{|w^\top a - \gamma|}{\|w\|_{p'}}$ is a convex function of $(w, \gamma)$ if and only if the following holds for every $(w_1, \gamma_1)$ and $(w_2, \gamma_2) \in \mathbb{R}^{n+1}$ and $\lambda \in [0, 1]$:

$$\lambda \frac{|w_1^\top a - \gamma_1|}{\|w_1\|_{p'}} + (1 - \lambda) \frac{|w_2^\top a - \gamma_2|}{\|w_2\|_{p'}} \geq$$
$$\frac{|(\lambda w_1 + (1 - \lambda) w_2)^\top a - (\lambda \gamma_1 + (1 - \lambda) \gamma_2)|}{\|\lambda w_1 + (1 - \lambda) w_2\|_{p'}}. \tag{5}$$

Let $a = (0, 0)$ and consider two hyperplanes of parameters $w_1 := (1, -\frac{1}{5}), \gamma_1 = 1$ and $w_2 := (-\frac{1}{5}, 1), \gamma_2 = 1$. Let $\gamma := \gamma_1 = \gamma_2$. Letting $\lambda = \frac{1}{2}$, Inequality (5) reads:

$$\frac{1}{2} \frac{1}{\sqrt[p']{1 + \left(\frac{1}{5}\right)^{p'}}} + \frac{1}{2} \frac{1}{\sqrt[p']{1 + \left(\frac{1}{5}\right)^{p'}}} \geq \frac{1}{\sqrt[p']{\left(\frac{2}{5}\right)^{p'} + \left(\frac{2}{5}\right)^{p'}}}, \tag{6}$$

or, equivalently:

$$\sqrt[p']{\left(\frac{2}{5}\right)^{p'} + \left(\frac{2}{5}\right)^{p'}} \geq \sqrt[p']{1 + \left(\frac{1}{5}\right)^{p'}}.$$

Taking both sides to the $p'$-th power, we have $2 \left(\frac{2}{5}\right)^{p'} \geq 1 + \left(\frac{1}{5}\right)^{p'}$. After moving 1 to the left-hand side and multiplying both sides by $5^{p'}$, we deduce $2 \cdot 2^{p'} - 1 \geq 5^{p'}$, which, if valid, implies $2 \cdot 2^{p'} > 2 \cdot 2^{p'} - 1 \geq 5^{p'}$. As $\left(\frac{5}{2}\right)^{p'} > 2$ holds for every $p' \in \mathbb{N} \cup \{\infty\}$ (as one can see by setting $p'$ to its smallest value, i.e., setting $p' := 1$), Inequality (6) is proven not to hold for any choice of $p \in \mathbb{N} \cup \{\infty\}$. $\square$

**Lemma 1.** *$k$-HC$_{(2,1)}$ and $k$-HC$_2$ coincide. Also, $k$-HC$_{(p,c)}$ is quadratically homogeneous w.r.t. $c$, i.e., $\mathrm{OPT}(k\text{-HC}_{(p,c)}) = c^2 \mathrm{OPT}(k\text{-HC}_{(p,1)})$.*

*Proof.* We start by showing that $k$-HC$_2^{\geq 1}$ and $k$-HC$_2$ are equivalent when $c = 1$ and $p = 2$. Indeed, as $n$ points in general position fix a hyperplane in $\mathbb{R}^n$, only $n$ of the $n + 1$ parameters in $(w_j, \gamma_j) \in \mathbb{R}^{n+1}$ are independent. Thus, $\|w_j\|_2^2 = \|w_j\|_2 = 1$ can be imposed w.l.o.g. for all $j \in [k]$. Relaxing $\|w_j\|_2 = 1$ as $\|w_j\|_2 \geq 1$ is w.l.o.g. as the latter is tight in any optimal solution—indeed, if not, a strictly better solution is found by scaling $(w_j, \gamma_j)$ by $\frac{1}{\|w_j\|_{p'}}, j \in [k]$. Let $\{(w_j, \gamma_j)\}_{j \in [k]}$ be an optimal solution to $k$-HC$_p^{\geq c}$. As argued, $\|w_j\|_{p'} = c$ holds. Let now $(w_j', \gamma_j') := \frac{(w_j, \gamma)}{c}, j \in [k]$. Such a scaled solution satisfies $\|w_j'\|_{p'} = 1$ for all $j \in [k]$ and, thus, is feasible for $k$-HC$_p^{\geq 1}$. Its objective function value is $\frac{1}{c^2}$ times the one of $\{(w_j, \gamma)\}_{j \in [k]}$. Since such a multiplicative difference is a constant, the scaled solution is optimal for $k$-HC$_p^{\geq 1}$. Thus, we have $\mathrm{OPT}(k\text{-HC}_p^{\geq c}) = c^2 \mathrm{OPT}(k\text{-HC}_p^{\geq 1})$. $\square$

**Theorem 1.** *Let $p, q \in \mathbb{N} \cup \{\infty\}$ and $c > 0$. The three positive scalars $\alpha(p, q), \beta(p, q), \gamma(p, q)$ which satisfy the congruence relationship*

$$\alpha(p, q)\|x\|_p \leq \beta(p, q)\|x\|_q \leq \gamma(p, q)\|x\|_p \qquad \forall x \in \mathbb{R}^n \tag{7}$$

*for $p, q \in \mathbb{N} \cup \{\infty\}$ also satisfy*

$$\frac{\alpha(p, q)^2}{\gamma(p, q)^2} \mathrm{OPT}(k\text{-HC}_{(p,c)}) \leq \mathrm{OPT}\left(k\text{-HC}_{(q, c\frac{\beta(p,q)}{\gamma(p,q)})}\right) \leq \mathrm{OPT}(k\text{-HC}_{(p,c)}). \tag{8}$$

*Proof.* The inequality

$$\min_{x \in X} f(x) \leq \min_{x \in X} f'(x) \leq \min_{x \in X} f''(x) \tag{9}$$

holds for any three functions $f, f', f'' : X \to \mathbb{R}$ satisfying $f(x) \leq f'(x) \leq f''(x)$ for all $x \in X \subseteq \mathbb{R}^n$. Since vector norms in $\mathbb{R}^n$ are congruent, for every $p, q \in \mathbb{N} \cup \{\infty\}$ there are three positive scalars $\alpha(p, q), \beta(p, q), \gamma(p, q)$ which satisfy equation 7. Since, by definition, $d_p(a, H) = \min_{y \in H} \|a - y\|_p$, equation 9 leads to the following congruence relationship for point-to-hyperplane distances that holds for every hyperplane $H$ in $\mathbb{R}^n$ and point $a \in \mathbb{R}^n$:

$$\alpha(p, q)\, d_p(a, H) \leq \beta(p, q) d_q(a, H) \leq \gamma(p, q)\, d_p(a, H). \tag{10}$$

Squaring equation 10 and letting $H_1, \ldots, H_k$ be an arbitrary choice of $k$ hyperplanes, another application of equation 9 leads to

$$\alpha(p, q)^2 \min_{j \in [k]} \{d^2(a_i, H_j)_p\} \leq \beta(p, q)^2 \min_{j \in [k]} \{d^2(a_i, H_j)_q\} \leq \gamma(p, q)^2 \min_{j \in [k]} \{d^2(a_i, H_j)_p\}. \tag{11}$$

Summing over the data points, we obtain the following surrogate inequality:

$$\alpha(p, q)^2 \sum_{i=1}^{m} \min_{j \in [k]} \{d^2(a_i, H_j)_p\} \leq \beta(p, q)^2 \sum_{i=1}^{m} \min_{j \in [k]} \{d^2(a_i, H_j)_q\} \leq \gamma(p, q)^2 \sum_{i=1}^{m} \min_{j \in [k]} \{d^2(a_i, H_j)_p\}.$$

Applying again equation 9 for the choice of the optimal hyperplane equations, we deduce $\alpha(p, q)^2 \operatorname{OPT}(k\text{-HC}_p{}^{\geq 1}) \leq \beta(p, q)^2 \operatorname{OPT}(k\text{-HC}_q{}^{\geq 1}) \leq \gamma(p, q)^2 \operatorname{OPT}(k\text{-HC}_p{}^{\geq 1})$. Multiplying through by $c^2$ and using Lemma 1, we obtain $\alpha(p, q)^2 \operatorname{OPT}(k\text{-HC}_p{}^{\geq c}) \leq \beta(p, q)^2 \operatorname{OPT}(k\text{-HC}_q{}^{\geq c}) \leq \gamma(p, q)^2 \operatorname{OPT}(k\text{-HC}_p{}^{\geq c})$. By using Lemma 1 one more time, we deduce $\beta(p, q)^2 \operatorname{OPT}(k\text{-HC}_q{}^{\geq c}) = \operatorname{OPT}(k\text{-HC}_q{}^{\geq c\beta(p,q)})$, which allows us to write:

$$\alpha(p, q)^2 \operatorname{OPT}(k\text{-HC}_p{}^{\geq c}) \leq \operatorname{OPT}(k\text{-HC}_q{}^{\geq c\beta(p,q)}) \leq \gamma(p, q)^2 \operatorname{OPT}(k\text{-HC}_p{}^{\geq c}).$$

Dividing through by $\gamma(p, q)$ and applying Lemma 1 one last time, the claim is obtained. □

**Corollary 1.** $k\text{-HC}_{(\infty, 1)}$ and $k\text{-HC}_{(1, \frac{1}{\sqrt{n}})}$ satisfy:

$$\frac{1}{n} \operatorname{OPT}(k\text{-HC}_{(2,1)}) \leq \operatorname{OPT}(k\text{-HC}_{(\infty,1)}) \leq \operatorname{OPT}(k\text{-HC}_{(2,1)})$$

$$\frac{1}{n} \operatorname{OPT}(k\text{-HC}_{(2,1)}) \leq \operatorname{OPT}(k\text{-HC}_{(1, \frac{1}{\sqrt{n}})}) \leq \operatorname{OPT}(k\text{-HC}_{(2,1)}).$$

*Proof.* We rely on the following congruence relationships:

$$\frac{1}{\sqrt{n}} \|x\|_2 \leq \|x\|_\infty \leq \|x\|_2 \qquad\qquad \frac{1}{\sqrt{n}} \|x\|_2 \leq \frac{1}{\sqrt{n}} \|x\|_1 \leq \|x\|_2.$$

Thanks to Theorem 1, $\frac{1}{\sqrt{n}} \|x\|_2 \leq \|x\|_\infty \leq \|x\|_2$ implies $\frac{1}{n} \operatorname{OPT}(k\text{-HC}_2{}^{\geq 1}) \leq \operatorname{OPT}(k\text{-HC}_\infty{}^{\geq 1}) \leq \operatorname{OPT}(k\text{-HC}_2{}^{\geq 1})$. Thanks to Theorem 1, $\frac{1}{\sqrt{n}} \|x\|_2 \leq \frac{1}{\sqrt{n}} \|x\|_1 \leq \|x\|_2$ implies $\frac{1}{n} \operatorname{OPT}(k\text{-HC}_2{}^{\geq 1}) \leq \frac{1}{n} \operatorname{OPT}(k\text{-HC}_1{}^{\geq 1}) \leq \operatorname{OPT}(k\text{-HC}_2{}^{\geq 1})$ which, due to Lemma 1, is equal to $\frac{1}{n} \operatorname{OPT}(k\text{-HC}_2{}^{\geq 1}) \leq \operatorname{OPT}(k\text{-HC}_1{}^{\geq \frac{1}{\sqrt{n}}}) \leq \operatorname{OPT}(k\text{-HC}_2{}^{\geq 1})$. □

**Lemma 2.** *Imposing* $\min\{\|w\|_1, \sqrt{n}\|w\|_\infty\} \geq 1$ *coincides with accounting for each point-to-hyperplane distance as* $\max\{d_\infty(a_i, H_j), \frac{1}{\sqrt{n}} d_1(a_i, H_j)\}$, *which translates in measuring the distance between* $a_i$ *and the closest point on* $H_j$, *call it* $y$, *as* $\max\{\|a_i - y\|_\infty, \frac{1}{\sqrt{n}}\|a_i - y\|_1\}$.

*Proof.* In the context of point-to-hyperplane distances, $\min\{\|w\|_1, \sqrt{n}\|w\|_\infty\} = 1$ implies $|a_i^\top w_j - \gamma| = \frac{|a_i^\top w_j - \gamma|}{\min\{\|w\|_1, \sqrt{n}\|w\|_\infty\}}$. We can rewrite the latter as $\max\{\frac{|a_i^\top w_j - \gamma|}{\|w\|_1}, \frac{|a_i^\top w_j - \gamma|}{\sqrt{n}\|w\|_\infty}\} = \max\{\frac{|a_i^\top w_j - \gamma|}{\|w\|_1}, \frac{1}{\sqrt{n}} \frac{|a_i^\top w_j - \gamma|}{\|w\|_\infty}\} = \max\{d_\infty(a_i, H_j), \frac{1}{\sqrt{n}} d_1(a_i, H_j)\}$. Such a multi orthogonal distance clearly stems from the norm $\max\{\|x\|_\infty, \frac{1}{\sqrt{n}}\|x\|_1\}$. □

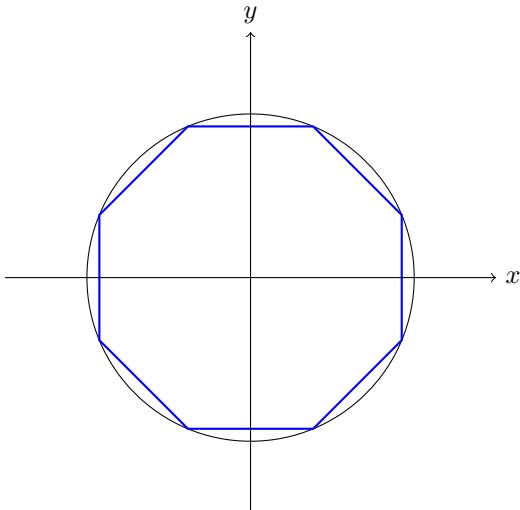

Figure 4: Sets of points satisfying $||x||_2 = 1$ (outer) and $\max\{||x||_\infty, \frac{1}{\sqrt{n}}||x||_1 = 1\}$ (inner).

**Lemma 3.** $\max\{||x||_\infty, \frac{1}{\sqrt{n}}||x||_1\}$ *is a norm and it satisfies the congruence relationship*

$$1 \Big/ \sqrt{1 + \frac{(\sqrt{n}-1)^2}{(n-1)}}||x||_2 \leq \max\{||x||_\infty, \frac{1}{\sqrt{n}}||x||_1\} \leq ||x||_2 \qquad \forall x \in \mathbb{R}^n.$$

*Proof.* Let us show that $\max\{||x||_\infty, \frac{1}{\sqrt{n}}||x||_1\}$ is a norm. First, it is clear that $\max\{||x||_\infty, \frac{1}{\sqrt{n}}||x||_1\} = 0$ if and only if $x = 0$. Second, it is also clear that $\lambda \max\{||x||_\infty, \frac{1}{\sqrt{n}}||x||_1\} = \max\{\lambda ||x||_\infty, \lambda \frac{1}{\sqrt{n}}||x||_1\}$. Third, we must show $\max\{||x + y||_\infty, \frac{1}{\sqrt{n}}||x+y||_1\} \leq \max\{||x||_\infty, \frac{1}{\sqrt{n}}||x||_1\} + \max\{||y||_\infty, \frac{1}{\sqrt{n}}||y||_1\}$. To see this, we first notice that

$$||x + y||_\infty \leq ||x||_\infty + ||y||_\infty$$
$$\frac{1}{\sqrt{n}}||x + y||_1 \leq \frac{1}{\sqrt{n}}||x||_1 + \frac{1}{\sqrt{n}}||y||_1$$

hold since these functions are norms. Taking the maximum of the left-hand and right-hand sides, we have:

$$\max\{||x + y||_\infty, \frac{1}{\sqrt{n}}||x+y||_1\} \leq \max\{||x||_\infty + ||y||_\infty, \frac{1}{\sqrt{n}}||x||_1 + \frac{1}{\sqrt{n}}||y||_1\}.$$

To show that this implies that the triangle inequality is satisfied, we show that, for any $a, b, c, d \geq 0$, we have $\max\{a+c, b+d\} \leq \max\{a, b\} + \max\{c, d\}$. Note that $a \leq \max\{a, b\}$, $b \leq \max\{a, b\}$, $c \leq \max\{c, d\}$, and $d \leq \max\{c, d\}$. Adding the inequalities, we have: $a + c \leq \max\{a, b\} + \max\{c, d\}$ and $b + d \leq \max\{a, b\} + \max\{c, d\}$. Taking the maximum of the left- and right-hand sides, we have proven the property we sought to prove.

We are now looking to prove a congruence of type

$$\alpha ||x||_2 \leq \beta \max\{||x||_\infty, \frac{1}{\sqrt{n}}||x||_1\} \leq \gamma ||x||_2$$

for some $\alpha, \beta, \gamma \geq 0$. We can split it as follows:

$$\alpha ||x||_2 \leq \beta \max\{||x||_\infty, \frac{1}{\sqrt{n}}||x||_1\} \Leftrightarrow \frac{||x||_2}{\max\{||x||_\infty, \frac{1}{\sqrt{n}}||x||_1\}} \leq \frac{\beta}{\alpha}$$

and

$$\beta \max\{||x||_\infty, \frac{1}{\sqrt{n}}||x||_1\} \leq \gamma ||x||_2 \Leftrightarrow \frac{\beta}{\gamma} \leq \frac{||x||_2}{\max\{||x||_\infty, \frac{1}{\sqrt{n}}||x||_1\}}.$$

Now, $\max\{||x||_\infty, \frac{1}{\sqrt{n}}||x||_1\}$ is a convex function (it is the maximum of two convex functions). Hence its level curves are convex—see Figure 4.

The maximum of $||x||_2$ over $\max\{||x||_\infty, \frac{1}{\sqrt{n}}||x||_1\} = 1$ is at the breakpoints of the border of the level curve of the latter where the two norms are both equal to 1, i.e., where $||x||_\infty = 1$ and $\frac{1}{\sqrt{n}}||x||_1 = 1$, i.e., $||x||_1 = \sqrt{n}$. A) We impose $x_1 = 1$. B) We impose $1 + \sum_{j=2}^n |x_j| = \sqrt{n}$ and assume (w.l.o.g.) $w \geq 0$, $1 + \sum_{j=2}^n x_j = \sqrt{n}$. C) We maximize $||x||_2$ by maximizing $1 + \sum_{j=2}^n x_j^2 : \sum_{j=2}^n w_j = \sqrt{n} - 1$. D) The Lagrangian function is: $\sum_{j=2}^n w_j^2 + \lambda(\sum_{j=2}^n w_j - \sqrt{n} + 1)$. E) The KKTs are: (i) $2w_j = -\lambda$ (gradient of the Lagrangian equal to 0) and (ii) $\sum_{j=2}^n w_j = \sqrt{n} - 1$ (primal constraint). F) From (i), we deduce $w_j = -\frac{1}{2}\lambda$. G) Plugging such a value into (ii), we obtain: $-(n-1)\frac{1}{2}\lambda = \sqrt{n} - 1$; this implies $\lambda = -2\frac{\sqrt{n}-1}{(n-1)}$. H) Thus, we have $w_j = \frac{\sqrt{n}-1}{(n-1)}$. I) In turn: $||w||_2 = \sqrt{1 + (n-1)\left(\frac{\sqrt{n}-1}{(n-1)}\right)^2} = \sqrt{1 + \frac{(\sqrt{n}-1)^2}{(n-1)}}$. Since this quantity is larger than 1, we have shown $\frac{||x||_2}{\max\{||x||_\infty, \frac{1}{\sqrt{n}}||x||_1\}} \leq \frac{\beta}{\alpha} = \sqrt{1 + \frac{(\sqrt{n}-1)^2}{(n-1)}}$. This implies $||x||_2 \leq \sqrt{1 + \frac{(\sqrt{n}-1)^2}{(n-1)}} \frac{||x||_2}{\max\{||x||_\infty, \frac{1}{\sqrt{n}}||x||_1\}}$. Since both $||w||_\infty \leq ||w||_2$ and $\frac{1}{\sqrt{n}}||w||_1 \leq ||w||_2$, we deduce $\max\{||x||_\infty, \frac{1}{\sqrt{n}}||x||_1\} \leq ||x||_2$ (which implies $1 = \frac{\beta}{\gamma} \leq \frac{||x||_2}{\max\{||x||_\infty, \frac{1}{\sqrt{n}}||x||_1\}}$). Combining the two, we have:

$$\max\{||x||_\infty, \frac{1}{\sqrt{n}}||x||_1\} \leq ||x||_2 \leq \sqrt{1 + \frac{(\sqrt{n}-1)^2}{(n-1)}} \max\{||x||_\infty, \frac{1}{\sqrt{n}}||x||_1\}. \quad (12)$$

Now, we multiply through by the inverse of the coefficient $\sqrt{1 + \frac{(\sqrt{n}-1)^2}{(n-1)}}$ and obtain:

$$\frac{1}{\sqrt{1 + \frac{(\sqrt{n}-1)^2}{(n-1)}}} \max\{||x||_\infty, \frac{1}{\sqrt{n}}||x||_1\} \leq \frac{1}{\sqrt{1 + \frac{(\sqrt{n}-1)^2}{(n-1)}}} ||x||_2 \leq \max\{||x||_\infty, \frac{1}{\sqrt{n}}||x||_1\}.$$

$$(13)$$

Combining the second part of (13) with the first part of (12), we obtain:

$$\frac{1}{\sqrt{1 + \frac{(\sqrt{n}-1)^2}{(n-1)}}} ||x||_2 \leq \max\{||x||_\infty, \frac{1}{\sqrt{n}}||x||_1\} \leq ||x||_2.$$

$\square$

**Corollary 2.** $k$-$\text{HC}_{(\text{multi},1)}$ *enjoys the following approximation relationship:*

$$1 \Big/ \left(1 + \frac{(\sqrt{n}-1)^2}{(n-1)}\right) \text{OPT}(k\text{-HC}_{(2,1)}) \leq \text{OPT}(k\text{-HC}_{(\text{multi},1)}) \leq \text{OPT}(k\text{-HC}_{(2,1)}).$$

*Proof.* A direct consequence of applying Theorem 1 to the congruence relationship derived in Lemma 3. $\square$

**Proposition 2.** *Under Assumption 1, when solving $k$-$\text{HC}_{(2,1)}$ a nonzero lower bound is obtained only after generating $\Omega(2^{k(n-1)})$ nodes.*

*Proof.* By assumption, each branching operation decides the sign of a component of $w_j$ for some $j \in [k]$ by splitting (with a half-space constraint) its feasible region with a hyperplane containing the origin. As long as the cone, call it $C$, obtained by intersecting such half-spaces is not pointed, the convex hull of its intersection with the feasible region of the problem contains the origin. Thus, the solution with $(w_j, \gamma_j) = 0$ and $x_{ij} = 1$, $i \in [m]$, which coincides with assigning every data point to

the degenerate hyperplane of index $j$ (thus achieving a $d_i = 0$, $i \in [m]$), is optimal regardless of the convex envelope that is employed. Only after branching has been carried out on each component of $w_j$ for each $j \in [k]$, the cone $C$ is pointed and, thus, the convex hull of its intersection with the feasible region of the problem renders the trivial solution $(w_j, \gamma_j) = 0$, $j \in [k]$, infeasible, leading to a nonzero lower bound. This amounts to generating $\Omega(2^{k(n-1)})$ nodes. $\square$

**Proposition 3.** *Assume that the constraint $\|w_j\|_1 \geq 1$, $j \in [k]$, is imposed and that branching takes place on the $s_{jh}$ variables first. Then, a nonzero global lower bound is calculated after generating $\Theta(2^{k(n-1)})$ nodes; after this, no further branching on $w$ takes place.*

*Proof.* Let $s_{jh} = \frac{1}{2}$ for all $h \in [n]$, which implies $w_{jh}^+ \leq \frac{1}{2}$ and $w_{jh}^- \leq \frac{1}{2}$. Letting $w_{jh}^+ = w_{jh}^- = \frac{1}{2}$, we have $w_{jh}^+ + w_{jh}^- = 1$. This feasible solution trivially satisfies the 1-norm constraint equation 3d with $w_{jh}^+ - w_{jh}^- = w_{jh} = 0$. Thus, $(w_j, \gamma_j) = 0$, $j \in [k]$, is optimal. By branching on a variable $s_{jh}$, we impose either $w_{jh} \leq 0$ (with $s_{jh} = 0$) or $w_{jh} \geq 0$ (with $s_{jh} = 1$). In both cases, the solution where $w_{jh}^+ = w_{jh}^- = \frac{1}{2}$ and $w_{jh} = 0$ becomes infeasible due either $w_{jh}^+$ or $w_{jh}^-$ being forced to 0, but the solution with $w_{jh'} = 0$, for any other $h' \in [n] \setminus \{h\}$, remains feasible as long as branching on it has not taken place. Thus, a nonzero lower bound is obtained only in $\Omega(2^{k(n-1)})$ nodes. When such an exponentially-large tree of depth $k(n-1)$ is complete, though, $\|w_j\|_1 \geq 1$, $j \in [k]$, holds in each leaf node and, thus, no further branching on $w$ is necessary. $\square$

**Proposition 4.** *Assume that $\|w_j\|_\infty \geq \frac{1}{\sqrt{n}}$, $j \in [k]$, is imposed and that branching takes place on the $u_{jh}$ variables first. Then, $O(nk)$ nodes suffice to obtain a nonzero lower bound; after this, no further branching on $w$ takes place.*

*Proof.* After branching on $u_{jh}$ for any pair $j, h$, the (left, w.l.o.g.) child node with $u_{jh} = 1$ satisfies $w_{jh} \geq \sqrt{n}$. This guarantees $\|w_j\|_\infty \geq \sqrt{n}$ and, thus, no further branching is needed on $w_j$ in the descendants of the left node. Further branching operations on $w_j$ are only necessary on the right child node where $u_{jh} = 0$ has been imposed. By iteratively applying this reasoning, we obtain a tree with exactly two nodes per level (except for the root node) where each left node satisfies the $\|w_j\|_\infty \geq \sqrt{n}$ constraint for at least a $j \in [k]$. Therefore, when the three has depth $nk$, $\|w_j\|_\infty \geq \sqrt{n}$ is satisfied for all $j \in [k]$. When such an polynomially-sized tree of depth $k(n-1)$ is complete, $\|w_j\|_\infty \geq \sqrt{n}$, $j \in [k]$, holds in each leaf node and, thus, no further branching on $w$ is necessary. $\square$

## C PROOF OF THE APPROXIMATION FACTORS AND OF THEIR TIGHTNESS

We will rely on the following Lemma:

**Lemma 4.** *Given two functions $f, g : \mathbb{R}^n \to \mathbb{R}$ with $g$ surjective we have:*

$$\max_{x \in \mathbb{R}^n} \frac{f(x)}{g(x)} = \max_{\nu \in \mathbb{R}} \left\{ \max_{x \in \mathbb{R}^n} \left\{ \frac{f(x)}{\nu} : g(x) = \nu \right\} \right\}. \tag{14}$$

*If, for all $x \in \mathbb{R}^n$, $f(x) = f(|x|)$ and $g(x) = g(|x|)$, then:*

$$\max_{x \in \mathbb{R}^n} \frac{f(x)}{g(x)} = \max_{\nu \in \mathbb{R}_+} \left\{ \max_{x \in \mathbb{R}_+^n} \left\{ \frac{f(x)}{\nu} : g(x) = \nu \right\} \right\}. \tag{15}$$

*Proof.* If $g$ is surjective, then $\cup_{\nu \in \mathbb{R}} \{x \in \mathbb{R}^n : g(x) = \nu\} = \mathbb{R}^n$. We can therefore partition $\mathbb{R}^n$ into infinitely many subsets of type $\{x \in \mathbb{R}^n : g(x) = \nu\}$. An optimal solution to $\max_{x \in \mathbb{R}^n} \frac{f(x)}{g(x)}$ thus corresponds to the best solution over all such subsets. The special case in Equation equation 15 follows by a similar argument. $\square$

**Proposition 5.** *The following relationships are satisfied for every $x \in \mathbb{R}^n$:*

$$\|x\|_2 \leq \|x\|_1 \leq \sqrt{n}\|x\|_2$$

$$\frac{1}{\sqrt{n}}\|x\|_2 \leq \|x\|_\infty \leq \|x\|_2$$

*and the factors $\sqrt{n}$ and $\frac{1}{\sqrt{n}}$ are tight.*

*Proof.* We are looking for four positive coefficients $\alpha_1, \beta_1, \alpha_\infty, \beta_\infty$ that satisfy the following relationships for all $x \in \mathbb{R}^n$:

$$\alpha_1 \|x\|_2 \leq \|x\|_1 \leq \beta_1 \|x\|_2$$
$$\alpha_\infty \|x\|_2 \leq \|x\|_\infty \leq \beta_\infty \|x\|_2.$$

Assuming $x \neq 0$ as, for $x = 0$, $\alpha\|x\|_p \leq \|x\|_q \leq \beta\|x\|_p$ holds for all $\alpha, \beta$ and for all $p, q \in \mathbb{N} \cup \{\infty\}$, the tightest values for $\alpha_1, \beta_1, \alpha_\infty, \beta_\infty$ must satisfy the following relationships:

$$\beta_1 = \max_{x \in \mathbb{R}^n} \frac{\|x\|_1}{\|x\|_2} \qquad\qquad \beta_\infty = \max_{x \in \mathbb{R}^n} \frac{\|x\|_\infty}{\|x\|_2}$$

$$\alpha_1 = \min_{x \in \mathbb{R}^n} \frac{\|x\|_1}{\|x\|_2} \qquad\qquad \alpha_\infty = \min_{x \in \mathbb{R}^n} \frac{\|x\|_\infty}{\|x\|_2}.$$

As $\max \frac{\|x\|_p}{\|x\|_q} = \min \frac{\|x\|_q}{\|x\|_p}$ holds for all $p, q \in \mathbb{N} \cup \{\infty\}$, we need to solve the following four problems:

$$\beta_1 = \max \frac{\|x\|_1}{\|x\|_2} \qquad\qquad \beta_\infty = \max \frac{\|x\|_\infty}{\|x\|_2}$$

$$\alpha_1 = \max \frac{\|x\|_2}{\|x\|_1} \qquad\qquad \alpha_\infty = \max \frac{\|x\|_2}{\|x\|_\infty}.$$

Let us consider the case of $\alpha_1, \alpha_\infty$, for which we are solving $\max \frac{\|x\|_2}{\|x\|_q}$ for $q = 1, \infty$. By virtue of Lemma 4, we are thus solving:

$$\alpha_q = \max_{\nu \in \mathbb{R}_+} \left\{ \frac{1}{\nu} \max_{x \in \mathbb{R}_+^n} \{ \|x\|_2 : \|x\|_q = \nu \} \right\}.$$

As the maximum of a convex function (such as $\|x\|_2$) over a closed, convex set is achieved on the border of the latter and, if we are optimizing over a polytope, over its extreme vertices, we can w.l.o.g. relax $\|x\|_q = \nu$ into $\|x\|_q \leq \nu$.

For $\alpha_1$, the extreme points of $\{ x \in \mathbb{R}^n : \|x\|_1 \leq \nu \}$ are of the form: $\nu e_\ell$ for all $\ell \in [n]$, with $e_\ell$ being the $\ell$-th canonical vector of $\mathbb{R}^n$. For each of them, we have $\|\nu e_\ell\|_2 = \sqrt{\nu^2} = \nu$. Thus, $\alpha_1 = \max \frac{\|x\|_2}{\|x\|_1} = \frac{\nu}{\nu} = 1$.

For $\alpha_\infty$, the extreme points of $\{ x \in \mathbb{R}^n : \|x\|_\infty \leq \nu \}$ are of the form: $(\pm\nu, \ldots, \pm\nu)$ for all possible choices of $\pm$. For each of them, we have $\|(\pm\nu, \ldots, \pm\nu)\|_2 = \sqrt{\nu^2 n} = \nu\sqrt{n}$. Thus, $\alpha_\infty = \max \frac{\|x\|_2}{\|x\|_\infty} = \frac{\nu\sqrt{n}}{\nu} = \sqrt{n}$.

Let us now consider the case of $\beta_1$ and $\beta_\infty$, for which we are solving $\max \frac{\|x\|_q}{\|x\|_2}$ for $q = 1, \infty$. By virtue of Lemma 4, we are thus solving:

$$\beta_q = \max_{\nu \in \mathbb{R}_+} \left\{ \frac{1}{\nu} \max_{x \in \mathbb{R}_+^n} \{ \|x\|_q : \|x\|_2 = \nu \} \right\}.$$

For $\beta_1$, the problem reads:

$$\beta_1 = \max_{\nu \geq 0} \left\{ \frac{1}{\nu} \max_{x \in \mathbb{R}_+^n} \{ e^T x : x^T x = \nu^2 \} \right\}. \tag{16}$$

The KKT conditions for the relaxation of the inner problem of equation 16 obtained after dropping the nonnegativity on $x$ read:

$$\nabla_x (e^T x - \lambda(x^T x - \nu^2)) = 0$$
$$x^T x = \nu^2,$$

with $\lambda$ unrestricted in sign. From the first equation, we deduce $x = \frac{e}{2\lambda}$. By substituting it in the second equation, we obtain $\frac{e^T e}{2^2 \lambda^2} = \nu^2$, that is, $\lambda = \frac{\sqrt{n}}{2\nu}$. Thus, we have $x = \frac{e}{\sqrt{n}}\nu$. Since the latter

is nonnegative, it is an optimal solution to both the relaxation of the inner problem of equation 16 with $x \in \mathbb{R}^n$ and its unrelaxed version with $x \in \mathbb{R}^n_+$. We thus have $\|x\|_1 = \frac{\nu}{\sqrt{n}} \|e\|_1 = \frac{\nu n}{\sqrt{n}} = \nu\sqrt{n}$.

We conclude that $\beta_1 = \frac{\nu\sqrt{n}}{\nu} = \sqrt{n}$.

For $\beta_\infty$, the problem reads:

$$\beta_\infty = \max_{\nu \geq 0} \left\{ \frac{1}{\nu} \max_{x \in \mathbb{R}^n_+} \left\{ \max_{\ell \in [n]} \{x_\ell\} : x^T x = \nu^2 \right\} \right\}.$$

The optimal solutions to the inner problem are of the form $\nu e_\ell$, where $e_\ell$ is a canonical vector of $\mathbb{R}^n$, for which we have $\|\nu e_\ell\|_\infty = \nu$. We conclude that $\beta_\infty = \frac{\nu}{\nu} = 1$. $\qquad\square$