# OpenReview forum: "Solving the 2-norm k-hyperplane clustering problem via multi-norm formulations"
_ICLR.cc/2025/Conference — Submitted to ICLR 2025_

### Official Review · Reviewer_29fM · 2024-10-28

**Soundness:** 2
**Presentation:** 2
**Contribution:** 2
**Rating:** 5
**Confidence:** 4

**Summary:**

This paper studies the hyperplane clustering problem using the L2-norm. To improve computational efficiency, the authors propose solving a generalized and a strengthened formulation utilizing other Lp-norms. All formulations of the clustering problem can be solved by the spatial branch-and-bound (SBB) method proposed by Amaldi & Coniglio (2013). When applied to the formulations proposed by the authors, the optimization time can be reduced by a factor of 1.5. Bounds on the optimal objective values between the classical and generalized formulation are derived.

**Strengths:**

The significant contribution of the paper is to propose a generalized formulation of the problem that can be solved more efficiently using SBB.

**Weaknesses:**

1, Results regarding clustering accuracy are relatively weak. The SBB algorithm can only find a lower bound, which can make the clustering results hard to interpret. Bounds between strengthened formulations and the original formulation are missing, for instance,  bound between $HC_{(2,1), (\infty, 1)}$ and $HC_{(2,1)}$ is not provided.

2, Corollary 2 might be wrong. The authors said that $1/(1+ \frac{(\sqrt{n} - 1)^2}{n-1})$ is strictly smaller than $1/n$ for all $n$. But $1/(1+ \frac{(\sqrt{n} - 1)^2}{n-1})$ converges to $1/2$, while $1/n$ converges to 0.

3, No applications using real datasets are provided.

4, Comparisons with existing methods are NOT included in the simulations. In addition, the following paper is highly relevant but is not discussed in sufficient detail: Hyperplane Clustering Via Dual Principal Component Pursuit by Manolis C. Tsakiris and Rene Vidal (ICML 2017). The following works are also related but are not mentioned in the paper: T. Zhang, A. Szlam, and G. Lerman, Median k-flats for hybrid linear modeling with many outliers, in Workshop on Subspace Methods, pages 234–241, 2009; G. Lerman, M. B. McCoy, J. A. Tropp, and T. Zhang, Robust computation of linear models by convex relaxation, Foundations of Computational Mathematics, 15(2):363–410, 2015. Please refer to the papers cited in "Hyperplane Clustering Via Dual Principal Component Pursuit" for more related work.

**Questions:**

1, How do these additional constraints affect the clustering results in terms of the hyperplanes? Can some theoretical guarantees be developed for recovering the true hyperplanes?

2, How does the proposed method compare with the methods proposed in "Hyperplane Clustering Via Dual Principal Component Pursuit" in terms of computational efficiency and clustering accuracy?

3, Can the error in Corollary 2 be fixed?

---

### Official Review · Reviewer_K2bf · 2024-11-02

**Soundness:** 3
**Presentation:** 3
**Contribution:** 3
**Rating:** 5
**Confidence:** 4

**Summary:**

This paper considers the k-hyperplane clustering problem where the goal is to find k hyperplanes to minimize the sum of squared distance from every data point to its closest hyperplane. This problem is computationally hard. The authors first show how to relax the $L_2$ constraint to $L_1$ and $L_{\infty}$ to transform the solution space into a polyhedra then they use the spatial branch-and-bound techniques to find a solution in the relaxed solution space. The experiments show that optimizing the relaxed formulation yields a speedup of tens of times that of solving the original formulation when running in the low-dimensional data set.

The problem studied is important and very hard. I think the ideas presented are interesting and useful. My concern is the algorithm does not scale as the dimension increases (as shown in the theoretical and experimental results).

**Strengths:**

1. Simple but useful techniques to tackle a very hard problem.
2. The experimental results show the techniques are useful in low-dimensional data sets.

**Weaknesses:**

1. The approximation ratio can be as large as $\Omega(n)$, by corollary 2 of the paper.
2. The experiment on the high-dimensional data set does not show a significant improvement. This phenomenon has been discussed in the paper. Proposition 3 shows that the branching nodes of the $L_1$ relaxation grow exponentially by the dimension $n$. Thus the results may not be very helpful in solving high dimensional clustering tasks.

**Questions:**

I see the formulations by only applying the $(1,\frac{1}{\sqrt{n}})$ or $(\infty,1)$ relaxtion already yields good speedup in the experiments. Why is it necessary to impose them both simultaneously?

---

### Official Review · Reviewer_Aumc · 2024-11-03

**Soundness:** 2
**Presentation:** 2
**Contribution:** 2
**Rating:** 5
**Confidence:** 4

**Summary:**

The paper considers the 2-norm k-hyperplane clustering problem. The problem is traditionally solved via Mixed Integer Quadratically Constrained Quadratic Programming (MI-QCQP), as reviewed in Section 4.2.

The main proposal is to strengthen this classic MI-QCQP formulation via adding extra norm constraints.

**Strengths:**

The proposed method seems to be faster than traditional approaches for certain instances, as shown in Tables 1 and 2.

**Weaknesses:**

The paper has a few major weaknesses.

First, the presentation is suboptimal with redundancies and unclarity. For example:
- The theorems and propositions in Section 3 appear to be natural consequences of the norm equivalence. Technically it is incremental. And it is unclear how this serves the subsequent sections (see below).
- At the end of the day, the paper proposes adding L1 and L-infinity constraints to the formulation, which already has L2 constraints. This leads to two questions. First, why is Section 3 useful? Second, these constraints are redundant as the L2 constraint $|| w_j ||_2\geq 1$ implies the L1 and L-infinity constraints, as the paper plots in Figures 1 and 2. While the paper means that the L1 and L-infinity constraints are useful during the execution of the SBB algorithm, this leads to great confusion. And furthermore, since the SBB algorithm is disconnected from the actual implementation (see the point below), and since there is a lack of formal descriptions of the SBB algorithm, it is hard to assess whether the proposed redundant constraints are useful for SBB.
- The discussions on spatial branch-and-bound in Section 4 are disconnected from the actual implementations of the methods. It appears to me that the algorithms are implemented by writing down the formulation and invoking Gurobi. Note that Gurobi is a commercial, closed-source solver. Its implementation is unlikely to be pure spatial branch-and-bound; if that were the case, and as per Proposition 2, the classic MI-QCQP formulation would take exponential time at every iteration of bound calculation. Tables 1 and 2 show, instead, that the classic MI-QCQP formulation could be fast for some instances. This leads to concerns about the correctness of Proposition 2 or whether it is obtained by considering a worst case. Indeed, having a nonzero lower bound is easy: just add a constraint $d_i \geq \epsilon$ for a sufficiently small epsilon>0. I would guess that doing so can also speed up the algorithm.
- The introduction of MI-QCQP and the proposed formulation is not clear. Specifically:
   - At Line 302, why is it that *The only nonconvexity of the formulation is due to the 2-norm constraints*? Isn't that $x_{ij}$ being binary is also a non-convex constraint?
   - Typo in Eq. (3f), and Eq. (3f) is not justified: Why do we need a binary variable $s_{jh}$ to bound the entries of $w_j$? Isn't it that we could just take $w_{jh}^+$ to be the positive entries of $w_{jh}$? This uniquely determines  $w_{jh}^+$ and  $w_{jh}^-$.
   - Constraints in Eq. (3e) are not justified: Why do we want every entry of $w_j$ is bounded in $[-1,1]$?

**Questions:**

See the above.

---

### Official Review · Reviewer_DvtT · 2024-11-03

**Soundness:** 3
**Presentation:** 2
**Contribution:** 2
**Rating:** 5
**Confidence:** 2

**Summary:**

The paper tackles the k-Hyperplane Clustering Problem (k-HC2), emphasizing the use of the 2-norm (Euclidean) distance. The goal in this problem is to identify k hyperplanes that minimize the total squared Euclidean distances between each data point and its nearest hyperplane. This task is particularly challenging due to the NP-hard nature of k-HC2; the computational complexity of fitting multiple points to hyperplanes while minimizing distances grows significantly, especially in higher-dimensional spaces.

**Strengths:**

- The authors present an innovative approach to solving k-HC2 by incorporating alternative norm constraints, specifically the 1-norm and ∞-norm, to develop a multi-norm formulation.

- Strengthening the k-HC2 formulation with 1-norm and ∞-norm constraints significantly accelerates the SBB algorithm, making it possible to solve instances that previously demanded extensive computation and achieving up to a 1.5 orders-of-magnitude improvement in some cases.

- The paper provides a rigorous theoretical foundation, including proofs and approximation relationships, for applying multiple norm constraints to tackle complex clustering problems.

**Weaknesses:**

The experimental results are somewhat unclear:

- The results appear to be based on simulations. Are there any findings from real-world datasets?

- The sample size is restricted under 100. What might be the reason for this limitation?

- Would it be possible to use plots for comparisons rather than tables?

Adding multiple norm constraints increases the complexity of the MI-QCQP formulation, which could make practical implementation more challenging. How would you address this issue?

**Questions:**

See weakness.

---

### Author Response · Authors · 2024-12-04

Dear all,

I wanted to apologize for the lack of interaction on my part, although I was hoping to be able to reply to the reviewers’ comments.

I am the only author of this paper and, tragically, a member of my family suddenly passed away a few weeks ago. Things have been rather hard lately.

I want to thank you all for reading my paper and giving me your opinion. I wish you all the best. Give a hug to your family members for me.

//The author

---

### Meta-Review · Area_Chair_3Z4R · 2024-12-23

**Metareview:**

This work modifies the spatial branch-and-bound (SBB) method for the 2-norm k-hyperplane clustering problem to achieve better computational efficiency. In particular, it is shown that the p-norm variants can provide meaningful lowerbounds and greatly reduce the number of SBB nodes. Presentation of the analysis, in particular the statement of Corollary 2, were questioned by the reviewers. Numerical experiments are also considered not very practical.

**Additional Comments On Reviewer Discussion:**

The author did not provide a rebuttal.

---

### Decision · Program_Chairs · 2025-01-22

Reject